# Noisy Data Pruning by Label Distribution Discrimination

## Abstract

Data pruning aims to prune large-scale datasets into concise subsets, thereby reducing computational costs during model training. While a variety of data pruning methods have been proposed, most focus on meticulously curated datasets, and relatively few studies address real-world datasets containing noisy labels. In this paper, we empirically analyze the shortcomings of previous gradient-based methods, revealing that geometry-based methods exhibit greater resilience to noisy labels. Consequently, we propose a novel two-stage noisy data pruning method that incorporates selection and re-labeling processes, which takes into account geometric neighboring information. Specifically, we utilize the distribution divergence between a given label and the predictions of its neighboring samples as an importance metric for data pruning. To ensure reliable neighboring predictions, we employ feature propagation and label propagation to refine these predictions effectively. Furthermore, we utilize re-labeling methods to correct selected subsets and consider the coverage of both easy and hard samples at different pruning rates. Extensive experiments demonstrate the effectiveness of the proposed method, not only on real-world benchmarks but also on synthetic datasets, highlighting its suitability for practical applications with noisy label scenarios.

## 1 Introduction

The explosive growth of datasets has been a pivotal factor driving the success of deep neural networks (DNNs) across various applications. However, training on large-scale datasets is not only time-consuming but also economically challenging Ho et al. (2020). In fact, a substantial portion of the training data is redundant, indicating that the excess data can be pruned without compromising model performance Marion et al. (2023). Consequently, considerable research efforts have been devoted to data pruning, employing various metrics to identify important samples, including loss Paul et al. (2021), distribution distance Xiao et al. (2024), uncertainty Coleman et al. (2020) and gradients Killamsetty et al. (2021b).

While these methods have been proven effective in their respective contexts, they often rely on the prior assumption that the data is perfectly labeled. For instance, Paul Paul et al. (2021) posits that samples with high loss values are hard samples that are essential to improve the model performance, while samples with low loss values are regarded as easy samples that can be pruned. However, when this assumption is violated, that is, the dataset contains mislabeled samples, the samples with larger gradient values may actually be the mislabeled ones. From a robustness perspective, the samples with small loss Lyu & Tsang (2019) will be more beneficial for enhancing the model robustness. Moreover, in real-world scenarios, data collection often involves complex processes such as crowd sourcing and web crawler, which may not conform to the assumption of perfectly labeled data.

Therefore, previous data pruning methods that operate under the assumption of perfectly labeled samples face two significant challenges. First, in the noisy label scenario, the prediction results of the model become inaccurate, leading to both noisy labels and hard samples generating outliers. Second, when the selected subset contains noisy samples, previous methods may cause the model to overfit to these noisy samples. As shown in Fig. 1, the performance of the loss-based method (GraNd) is significantly degraded compared to the geometry-based method (KCenter Sener & Savarese (2017)) in the noisy label scenario. This performance drop is attributed to the sample selection bias inherent in GraNd, which tends to identify mislabeled samples as hard samples. In

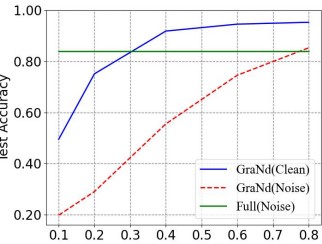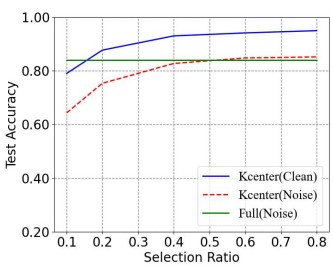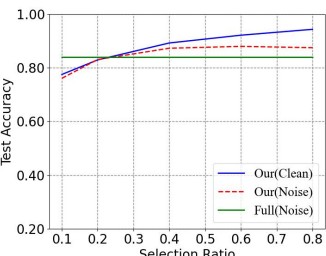

Figure 1: The different data pruning methods on clean label (CIFAR-10) and noisy label dataset (CIFAR-10N) at different pruning rates. Full means using the entire noisy label dataset.

contrast, KCenter considers the relationship among neighboring samples, resulting in less degradation in performance. Moreover, the model is prone to overfitting to noisy labels under noisy labels, leading to overall lower performance than methods designed for clean labels.

To solve these issues, an intuitive method is to find as many clean samples as possible from noisy samples to reduce sample selection bias, and then re-label the selected subset to prevent overfitting to the noisy samples. For instance, Adacore Pooladzandi et al. (2022) leverages the second-order information through the hessian matrix to minimize sample selection bias. Pr4ReL Park et al. (2024) uses robust learning methods to relabel the selected subset, maximizing the re-labeling accuracy and alleviating model overfitting. Unfortunately, these methods often fail to adequately balance the interplay between selection bias and the difficulty of sample re-labeling, which limits the overall effectiveness of the samples selected for re-labeling.

In this paper, we propose a two-stage **Ro**bust **P**runing (RoP) method, called RoP, which aims to effectively address these challenges by selecting and re-labeling. Firstly, we use the neighboring label inconsistency score (NLI-Score) to identify noisy label samples and select clean samples. Specifically, we assess the label distribution divergence between a given sample and its neighboring predictions to obtain the NLI-Score. To enhance the accuracy of neighboring predictions during obtaining NLI-Score, we employ feature propagation and label propagation techniques to refine these predictions. Second, we use robust learning methods to re-label the selected samples to prevent overfitting. Meanwhile, we empirically analyzing the difficulty of samples being re-labeled. Then, we leverage the density-based pruning method to ensure the coverage of easy and hard samples, thereby ensuring the benefits of subsets at different pruning rates. Our contributions are as follows:

- We propose a robust two-stage data pruning method by selecting and re-labeling, which first selects as many clean samples as possible by NLI-Score, and then re-labels the selected samples to avoid overfitting to noisy labels.

- We propose feature and label propagation to rectify the neighboring predictions during the NLI-Score estimation process, which improves the ability to identify noisy samples.

- Extensive experiments show that the proposed method is effective not only on synthetic noisy datasets but also on real-world benchmarks.

## 2 RELATED WORK

### 2.1 DATASET PRUNING

Existing data pruning methods can be roughly divided into two categories: score-based and optimization-based. Score-based methods usually select samples by carefully designed metrics based on gradients Paul et al. (2021); Zhang et al. (2024), feature embeddings Coleman et al. (2020), and model predictions Coleman et al. (2020), etc. For example, GraNd Paul et al. (2021) tends to select samples with large gradients, considering these samples as hard samples in the training process. However, it is not suitable in real noisy scenarios, as noisy samples also exhibit large gradients. Uncertainty Coleman et al. (2020) tends to select samples that the model is not confident about, as these samples may contain more information. K-Center Sener & Savarese (2017) removes redundant samples based on the similarity of samples in the feature space. Optimization-based methods

Killamsetty et al. (2021a); Xiao et al. (2024) aims to reduce the distribution bias between subset and the entire dataset. Some optimization-based methods ensure the distribution of the selected subset is close to the complete dataset by matching the gradients or feature distribution. Although optimization-based methods have theoretical guarantees, the data pruning process is usually very time-consuming.

Furthermore, recent studies Xia et al. (2022); Zheng et al. (2022a) have shown that existing socre-based methods do not work well at high pruning rates, which is due to neglecting the trade-off between the number of easy and hard samples. To mitigate this issue, some methods have designed different sampling strategy to ensure the coverage of easy and hard samples in extreme pruning rates. Although above methods have improved the robustness of data pruning under different pruning rates, they are still powerless against the selected samples with noisy labels.

## 2.2 NOISE LABEL LEARNING

Noisy label learning (NLL) Song et al. (2022); Li et al. (2022); Zhang et al. (2023) has emerged as an effective method to improve the robustness of DNNs, attracting widespread research interests. Early methods adopted a more direct insight, first aiming to identify clean samples by using carefully designed metric, and then reduce de-weighting potentially noisy samples during training, or leverage semi-supervised learning to re-relabel the noisy samples. For instance, Small Lyu & Tsang (2019) selects clean samples with small losses, combines them with the Gaussian Mixture Model (GMM) Reynolds et al. (2009) to estimate the noisy samples, and then reduces the loss weights of the noisy samples during the training process. Recently NLL studies Li et al. (2020); Liu et al. (2020; 2022a) have focused on leveraging self-consistency regularization to re-label the noisy samples. The mainly idea is to apply a series of strong augmentations to the input images and then use the consistency regularization loss to ensure the consistency between the augmented samples and the original ones. Popular methods in this family include DivideMix Li et al. (2020), ELR+ Liu et al. (2020), and SOP+ Liu et al. (2022a). DivideMix further improves the re-labeling accuracy by using a co-training framework, while SOP+ introduces additional learnable variables with a self-consistent loss. While these methods are highly effective, the re-labeling models often require more computational time and longer training epochs due to the additional data augmentations and multiple backbones, highlighting the need to improve their efficiency.

## 3 METHODOLOGY

In this section, we describe the preliminaries and show in detail how RoP can be applied in the noise label scenario. Fig. 2 presents an overview of RoP method, which comprises two stages, the first stage focuses on identifying noise samples, while the second stage is dedicated to re-labeling the pruned subsets.

## 3.1 PRELIMINARIES

We focus on data pruning for classification task, which is a widely studied scenario in machine learning community. We are given a training set $\tilde{\mathcal{D}} = \{(x_i, \tilde{y}_i)\}_{i=1}^n$ and a subset $\mathcal{S} = \{(x_i, \tilde{y}_i)\}_{i=1}^m$, where $x_i \in \mathcal{X}$, $\tilde{y}_i \in \tilde{\mathcal{Y}}$ denotes the label-sample pair (contains noisy label), $m$ and $n$ denotes the number of samples. Data pruning aims to find the most representative subset $\mathcal{S}^*$ from $\tilde{\mathcal{D}}$, so that the model $\theta_{\mathcal{S}^*}$ trained on $\mathcal{S}^*$ has closer generalization performance of the model $\theta_{\tilde{\mathcal{D}}}$ trained on the entire dataset $\tilde{\mathcal{D}}$.

## 3.2 NOISE LABEL DISCRIMINATION

Most methods for addressing noise label establish a selection criteria for noisy samples based on predicting the label distribution of individual samples Lyu & Tsang (2019). These methods often face challenges in avoiding selection bias, as they rely heavily on the evaluation of a single sample without considering neighboring relationships. In weakly supervised learning Zhou et al. (2003), there is the key prior assumption of consistency, which means points on the same structure (typically referred to as a cluster or a manifold) are likely to have the same label. From this perspective, candidate samples that may have clean labels can be found by evaluating the neighboring label

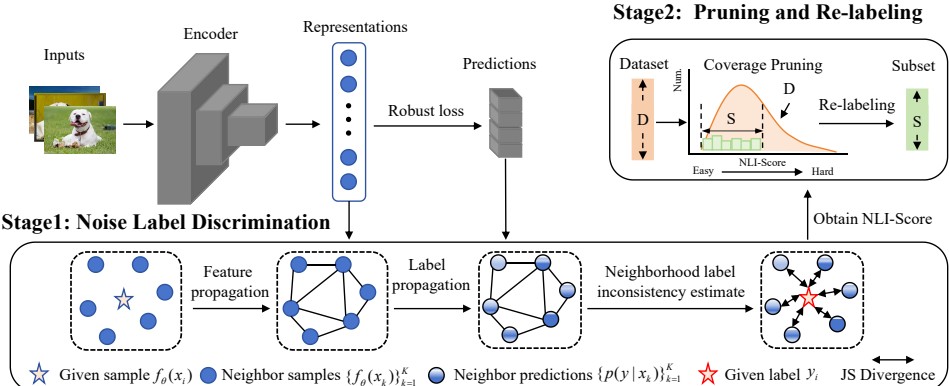

Figure 2: Illustration of the proposed framework. The proposed method mainly consists of two stages: noise label discrimination, pruning and re-labeling. The first stage aims to find as many clean samples as possible, which mainly includes three parts: feature propagation, label propagation, and neighborhood label inconsistency estimation. Specifically, we construct local graphs by propagation and label propagation to correct the neighboring predictions. Then, we use the label distribution divergence between the given label and its neighboring predictions to identify noisy samples. In the second stage, we use NLI-Score to select samples and re-label the selected subset by robust learning methods. During data pruning, we further balance the coverage of easy and hard samples by density-based sampling.

consistency of a given sample Iscen et al. (2022); Li et al. (2022). Therefore, we discriminate noisy samples by neighboring label consistency in the robust data pruning. Specifically, this process includes feature propagation, label propagation and neighboring label inconsistency estimation.

### 3.2.1 FEATURE PROPAGATION

As considering the neighboring label distribution, we first find the $K$ nearest neighbors of the given sample $x_i$ based on the cosine similarity in feature embedding space. Specifically, given a candidate sample-label pair $(x_i, \tilde{y}_i) \in \tilde{\mathcal{D}}_{train}$ ($\tilde{\mathcal{D}}_{train}$ contains noisy label), we use the cosine similarity in Eq. (1) to find the $K$ nearest neighbors of sample $x_i$.

$$cos(f_\theta(x_i), f_\theta(x_j)) = \frac{f_\theta^T(x_i) f_\theta(x_j)}{||f_\theta(x_i)||_2 ||f_\theta(x_j)||_2} \tag{1}$$

where $f_\theta(x_i)$ is the feature embedding of the sample $x_i$ from the pre-trained model $\theta_{\tilde{\mathcal{D}}}$ and $|| \quad ||_2$ denotes the $l_2$ regularization. And then, we define them as local neighboring samples (LNS), as formulated below.

$$\{x_k\}_{k=1}^K \leftarrow KNN(x_i; \tilde{\mathcal{D}}_{train}; K) \tag{2}$$

where $\{x_k\}_{k=1}^K$ defines as the LNS, $KNN(x_i; \tilde{\mathcal{D}}_{train}; K)$ is a function that returns $K$ most similar samples in $\tilde{\mathcal{D}}_{train}$ for the candidate sample $x_i$. Note that $x_i$ is temporarily removed from $\tilde{\mathcal{D}}_{trian}$ at this moment.

To further exploit the relationship among LNS, we use the $K$ nearest neighboring samples to define a local graph $G_{x_i}(V, E)$, where vertices matrix $V \in \mathbb{R}^{K \times d}$ contains the stacked neighboring features $\{f_\theta(x_k)\}_{k=1}^K$. To define the adjacency matrix $E \in \mathbb{R}^{K \times K}$, we first obtain the similarity matrix $S$, as formulated below.

$$S[i,j] = \begin{cases} cos(V[i,:], V[j,:]) & if \quad i \neq j \\ 0 & otherwise \end{cases} \tag{3}$$

where $V[i,:]$ denotes the i-th row of the matrix $V$. Note that in all backbone architectures utilized in our experiments, the penultimate layers are activated using a ReLu function, ensuring that all coefficients in $V$ are non-negative. Consequently, this implies that the coefficients in $S$ are also

non-negative and $S$ is symmetric. Then, we apply normalization to the resulting matrix.

$$E = D^{-1/2}SD^{-1/2}, \quad D[i,i] = \sum_j S[i,j] \tag{4}$$

where $E$ is the Laplacian of the adjacency matrix and $D$ is the degree diagonal matrix. Therefore, the graph vertices represent the neighbor features of sample $x_i$. Its nonzero weights are based on the cosine similarity between corresponding transferred representations.

We then apply feature propagation to obtain new features for each vertex, as formulated below.

$$V_{new} = (I + E)V \tag{5}$$

where $I$ denotes the identity matrix. $V_{new}$ represents the result of feature propagation among the $K$ neighboring samples.

### 3.2.2 LABEL PROPAGATION

After feature propagation, we use the $V_{new}$ to the fully connected (FC) layer of the model for label propagation, which aims to correct the predictions $\{p(y|x_k)\}_{k=1}^K$ of the neighboring samples $\{x_k\}_{k=1}^K$. The specific formula is as follows:

$$\{p(y|x_k)\}_{k=1}^K = softmax(V_{new}W_n) \tag{6}$$

where $W_n$ denotes the parameters of FC layer trained on noisy label dataset $\tilde{D}_{train}$. $p(y|x_k)$ denotes the predicted output of the mode on the sample $x_k$.

### 3.2.3 NEIGHBORHOOD LABEL INCONSISTENCY ESTIMATE

In order to reduce selection bias, instead of directly using model predictions on a given sample $x_i$ to identify the noisy label, we consider the consistency of its nerghborhood samples. The neighboring label inconsistency score between the given sample $x_i$ and neighboring predictions $\{p(y|x_k)\}_{k=1}^K$ can be defined as follow.

$$N_{score}(x_i, y_i) = \frac{1}{K}\sum_{k=1}^K JS(p(y|x_k), y_i) \tag{7}$$

where $N_{score}(x_i, y_i)$ denotes the NLI-Score value of the sample $x_i$, $y_i$ is the one-hot vector for the given ground-truth label of the sample $x_i$. And, $p(y|x_k)$ denotes the predicted probability of the $k$-th neighbor sample. JS denotes Jensen-Shannon divergence, as formulated follow:

$$JS(p_i, p_j) = \frac{1}{2}KL(p_i||\frac{p_i + p_j}{2}) + \frac{1}{2}KL(p_j||\frac{p_i + p_j}{2}) \tag{8}$$

where $KL(||)$ represents the Kullback-Leibler (KL) divergence. $p_i$ and $p_j$ denotes the probability distribution of two different samples, respectively.

### 3.3 PRUNING AND RE-LABELING

According to the obtained NLI-score, we can distinguish the clean and noisy label samples as much as possible. When a candidate sample exhibits a small NLI-score, it indicates a strong consistency with the predicted labels of its neighboring samples. Consequently, samples with smaller NLI-score are more likely to be clean label samples. We can then utilize the NLI-score for data pruning, aiming to minimize the presence of noisy samples within subset. However, selection bias is an inherent challenge in the noise label scenarios. To address this, a viable approach is to employ re-labeling methods to correct the labels of pruned subsets.

### 3.3.1 RE-LABELING AND EMPIRICAL ANALYSIS

In this section, we use the SOTA noisy label learning methods to re-labeling the pruned subsets. In noisy label learning, re-labeling methods Song et al. (2019) have achieved SOTA generalization by

designing self-correction modules such as self-supervised regularization Liu et al. (2022b). For example, SOP+ Liu et al. (2022b) almost reaches the same performance on CIFAR-10N noisy dataset as on CIFAR-10 clean dataset. Therefore, we consider using SOP+ to re-labeling the pruned subsets.

Intuitively, samples with high neighboring consistency are more likely to be rectified. Therefore, we empirically analyze the correlation between the NLI-Score and the re-labeling accuracy by using SOP+ as the re-labeling method on CIFAR-10N in Fig. 3.

Specifically, we train SOP+ in the 20% randomly selected subset $S$ for a warm-up training with 10 epochs and calculate the NLI-Score for the entire training samples. Next, we fully train SOP+ on the random subset $S$. Last, we divide the entire training set into 14 bins according to the obtained NLI-Score and verify the average re-labeling accuracy for each bin. Fig. 3 shows that a strong correlation between the re-labeling accuracy and the neighboring label inconsistency score, and the relabeling accuracy decreases with the increase of the NLI-score.

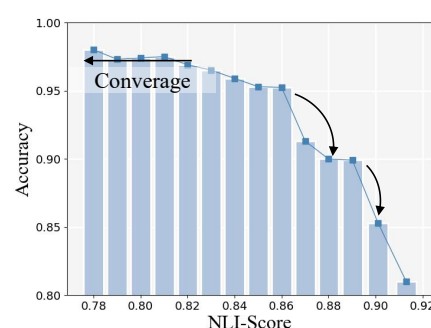

Figure 3: Empirical analysis about the relationship between the re-labeling accuracy and the NLI-Score values on CIFAR-10N.

Therefore, samples with lower NLI-Scores are more likely to be correctly re-labeled, while samples with higher NLI-Scores are more difficult to accurately annotate. Therefore, we identify samples with relatively low NLI-Scores as **easy samples** and those with high NLI-Scores as **hard samples**.

### 3.3.2 DENSITY-BASED COVERAGE PRUNING

Recent studies Toneva et al. (2018); Lyu & Tsang (2019) have shown that models tend to learn from easy samples before progressing to hard ones. However, hard samples are more effective to improve the generalization ability of the model. Previous score-based data pruning methods Paul et al. (2021); Coleman et al. (2020) have primarily focused on identifying and retaining hard samples, while neglecting the coverage of easy samples. This oversight has resulted in a significant decline in performance at high pruning rates.

Furthermore, methods such as Moderate Xia et al. (2022) and Small Lyu & Tsang (2019) indicate that retaining retaining as many easy samples as possible under high pruning is essential for maximizing the benefits. Therefore, we advocate for a coverage method that ensures the inclusion of both easy and hard samples at different pruning rates.

---

**Algorithm 1** RoP: Robust Pruning

**Input**: Training data $\tilde{\mathcal{D}} = \{x_i, \tilde{y}_i\}_{i=1}^n$, budget size $k$,
feature extractor $f_\theta$
**Training Variables**: Feature extractor $f_\theta$ pre-trained on $\tilde{\mathcal{D}}$
**Output**:Subset $\mathcal{S}^*$

1: Warm-up the feature extractor $f_\theta$ on $\tilde{\mathcal{D}}$
2: **for** $(x_i, y_i) \in \tilde{\mathcal{D}}$ **do**
3:    Feature propagation by Eq. (3)
4:    Label propagation by Eq. (6)
5:    Calculate NLI-Socre $E_{ver}(x_i, y_i^{(c)})$ by Eq. (7)
6: **end for**
7: Density-based coverage pruning $\mathcal{S}^*$
8: Re-labeling the subset $\mathcal{S}^*$
9: **Return** $\mathcal{S}^*$

---

Specifically, we use NLI-Score as an indicator to assess the difficulty of each sample in $\tilde{D}$. Subsequently, we apply the coverage coreset sampling (CCS) Zheng et al. (2022b) to ensure adequate representation of both hard and easy samples based on their NLI-Scores. CCS aims to strike a balance between the number of easy and hard samples, thereby alleviating the performance degradation often observed in high pruning rates. The complete algorithm of the Robust Pruning (RoP) method is outlined in Algorithm 1, which mainly contains the following five steps :

- Step 1 : Using feature propagation to update the neighboring features;
- Step 2 : Using label propagation to update the predictions of neighboring samples;
- Step 3 : Calculating the neighboring label inconsistency score of given samples;

- Step 4 : Obtaining the pruned subset $\mathcal{S}^*$ based on coverage pruning.
- Step 5 : Re-labeling the pruned subset $\mathcal{S}^*$ based on robust learning methods.

## 4 EXPERIMENTS

### 4.1 EXPERIMENTAL DATASET

In experiments, we mainly conduct experiments on four datasets, including three real noisy label datasets and one synthetic noisy label dataset. For real noisy label, we use CIFAR-10N, CIFAR-100N Wei et al. (2022), and Webvision Li et al. (2017). For synthetic noisy label, we use ImageNet-1K Deng et al. (2009) with asymmetric label noise. CIFAR-10N and CIFAR-100N contain human re-annotations of 50K training images in the original CIFAR-10 and CIFAR-100. Specifically, CIFAR-10N contain 3 random noisy labels, called Random 1,2,3, which are further transformed into the worst-case label. CIFAR-100N contains one type of noisy label, called Random. WebVision is a large-scale noisy datasets, which contains 2.4M images crawled from the Web using the 1k concepts in ImageNet-1K. Following prior work Chen et al. (2019), we use mini-WebVision consisting of the first 50 classes of the Google image subset with approximately 66K training images. Finally, we use ImageNet-1k to synthesize a asymmetric label noise dataset, called ImageNet-N, consisting of 1.2M training images.

### 4.2 PRUNING RATE

For CIFAR-10N, CIFAR-100N, and WebVision, we select the subset at the pruning rates 0.2, 0.4, 0.6, 0.8. For ImageNet-N, we select the subset at pruning rates 0.05, 0.1, 0.2, 0.4. We evaluate the accuracy of the selected subset by using SOP+ as the Re-labeling model. Every experiment is run 3 times, and the average of the last accuracy is reported. For CIFAR-10N with the random noise, we average the test accuracy of the models trained using the three noisy labels.

### 4.3 BASELINES

To demonstrate the effectiveness of the proposed method, we compare random selection and 10 data pruning methods, which are Small Lyu & Tsang (2019), Margin Coleman et al. (2020) , KCenter Sener & Savarese (2017) , Forget Toneva et al. (2018), GraNd Paul et al. (2021), SSP Sorscher et al. (2022), Moderate Xia et al. (2022), FDMat Xiao et al. (2024) and Pr4ReL Park et al. (2024). (1) KCenter selects $K$ samples as representative subsets whose maximum distance between samples is able to cover the entire training dataset. (2) GraNd takes the average norm of the gradient vector as a measure of the contribution of sample to the model. (3) Moderate aims to select samples with moderate difficulty to form a representative subset by selecting samples with median distance to the prototype. (4) FDMat aims to select a constituent representative subset whose feature distribution is close to the class prototype feature distribution. (5) Forgetting selects the samples that are easily forgotten (misclassified) by the classifier during the whole training process as representative samples. (6) Pr4Rel finds representative subsets by maximizing the re-labeling accuracy. (7) Margin selects samples by taking the difference between the first and second largest predicted probability of the model as a criterion to measure the difficulty of the sample. (8) SSP utilizes a self-supervised pre-trained model to select the most typical samples. (9) Small selects samples with small losses as typical samples. (10) Uniform selects samples with random selection.

### 4.4 EXPERIMENTS ON REAL NOISY DATASETS

We conduct experiments with real noisy labels on CIFAR10-N, CIFAR100-N and WebVision respectively. In Tab. 1, we compare RoP with 10 baseline methods on CIFAR-10N and CIFAR-100N, respectively. Overall, our method shows improved performance than the SOTA data pruning methods in the noisy label scenario. In particular, for the worst-case label scenario on CIFAR-10N, $ROP_B$ significantly outperforms the sub-optimal method, e.g., 1.5% improvement at 20% pruning rate. In addition, we also find that previous data pruning methods are indeed less robust in noisy label scenarios, for example GraNd has only 15.4% precision in worst-case label scenario of CIFAR-10N. This is because methods such as GraNd and Margin tend to prefer hard samples in the case of

Table 1: Comparison of baselines and the proposed RoP by using PreAct ResNet-18 and re-labeling method (SOP+) on CIFAR-10N and CIFAR-100N. RoP and $RoP_B$ denote pruning by NLI-Score without and with considering coverage, respectively. The best results are in bold.

| Re-label Methods | Selection Methods | CIFAR-10 | | | | | | | | CIFAR-100 | | | |
|---|---|---|---|---|---|---|---|---|---|---|---|---|---|
| | | Random(Noise ration ≈20%) | | | | Worst(Noise ratio≈40%) | | | | Random(Noise ratio≈40%) | | | |
| | | 0.2 | 0.4 | 0.6 | 0.8 | 0.2 | 0.4 | 0.6 | 0.8 | 0.2 | 0.4 | 0.6 | 0.8 |
| SOP+ | Random | 87.5±0.3 | 91.5±0.1 | 93.4±0.0 | 94.8±0.2 | 81.9±0.1 | 87.5±0.1 | 90.8±0.1 | 91.8±0.1 | 46.5±0.0 | 55.7±0.2 | 60.8±0.3 | 64.4±0.2 |
| | Small | 77.6±2.5 | 86.2±0.1 | 90.7±0.6 | 94.3±0.2 | 78.8±0.2 | 84.1±0.1 | 89.3±0.1 | 92.3±0.2 | 48.5±0.8 | 59.8±0.4 | 63.9±0.2 | 66.1±0.6 |
| | Margin | 52.1±5.0 | 79.6±8.6 | 92.6±3.9 | 95.1±1.3 | 45.7±1.1 | 61.8±0.7 | 84.6±0.3 | 92.5±0.0 | 20.0±1.2 | 34.4±0.3 | 50.4±0.6 | 63.3±0.1 |
| | KCenter | 86.3±0.4 | 92.2±0.3 | 94.1±0.2 | 95.3±0.1 | 81.9±0.0 | 88.0±0.0 | 91.3±0.1 | 92.3±0.0 | 44.8±0.6 | 55.9±0.3 | 61.6±0.3 | 65.2±0.6 |
| | Forget | 82.4±1.0 | 93.0±0.2 | 94.2±0.3 | 95.0±0.1 | 71.1±0.4 | 87.7±0.1 | 90.6±0.3 | 92.2±0.0 | 38.0±0.5 | 55.3±0.2 | 63.2±0.1 | 65.8±0.4 |
| | GraNd | 24.2±5.5 | 51.6±3.2 | 85.9±1.2 | 94.9±0.2 | 15.4±1.6 | 25.7±0.8 | 51.0±0.5 | 86.8±0.5 | 11.0±0.1 | 19.0±0.6 | 38.7±0.5 | 62.1±0.5 |
| | SSP | 80.5±2.6 | 91.7±1.5 | 93.8±1.0 | 95.0±0.2 | 70.8±2.7 | 86.6±1.9 | 89.2±0.9 | 92.3±0.4 | 39.2±2.2 | 54.9±1.5 | 62.7±0.7 | 65.0±0.3 |
| | Moderate | 87.8±1.0 | 92.8±0.5 | 94.0±0.3 | 94.9±0.2 | 75.2±1.5 | 81.9±1.2 | 87.7±0.7 | 91.8±0.3 | 46.4±1.8 | 54.6±1.7 | 60.2±0.4 | 64.6±0.4 |
| | Pr4ReL | 88.5±0.3 | 93.1±0.2 | 94.4±0.1 | 95.3±0.1 | 84.9±0.6 | 89.2±0.6 | 91.3±0.3 | 92.9±0.1 | 52.9±0.8 | 60.1±0.6 | 64.1±0.4 | 66.2±0.3 |
| | FDMat | 85.7±2.1 | 92.3±0.5 | 94.1±0.4 | 94.6±0.3 | 85.0±0.3 | 86.7±1.2 | 90.2±1.6 | 92.9±0.2 | 38.7±0.2 | 53.8±0.2 | 61.3±0.2 | 65.4±0.2 |
| | RoP | 87.9±-1.3 | 92.0±0.4 | 94.0±0.5 | 94.8±0.4 | **86.4±0.4** | **90.8±0.1** | **92.4±0.1** | **93.3±0.2** | **54.9±0.2** | 60.6±0.3 | 64.1±0.1 | 66.2±0.2 |
| | $RoP_B$ | **89.2±2.1** | **93.3±0.5** | **94.6±0.4** | 95.3±0.3 | 85.4±0.4 | 89.6±0.4 | 91.7±0.1 | **93.3±0.2** | 53.3±0.2 | **60.9±0.2** | **64.3±0.1** | **66.3±0.2** |

Table 2: Performance without re-labeling on CIFAR-10N and CIFAR-100N by using RreAct ResNet-18 over 3 different random seeds. The best results are in bold.

| Learning Models | Selection Methods | CIFAR-10 | | | | | | | | CIFAR-100 | | | |
|---|---|---|---|---|---|---|---|---|---|---|---|---|---|
| | | Random (Noise ratio≈20%) | | | | Worst(Noise ratio≈40%) | | | | Random(Noise ratio≈40%) | | | |
| | | 0.2 | 0.4 | 0.6 | 0.8 | 0.2 | 0.4 | 0.6 | 0.8 | 0.2 | 0.4 | 0.6 | 0.8 |
| CE | Random | 75.5±1.1 | 81.0±0.9 | 83.8±0.4 | 84.9±0.3 | 58.3±0.6 | 70.1±0.4 | 74.2±0.4 | 77.1±0.3 | 37.6±1.2 | 46.5±0.8 | 50.0±0.5 | 52.0±0.4 |
| | GraNd | 29.1±3.2 | 51.5±1.3 | 74.6±0.7 | 85.3±0.7 | 14.2±1.2 | 25.5±0.6 | 41.6±0.5 | 68.6±0.7 | 12.9±0.5 | 24.8±0.5 | 37.2±1.4 | 48.2±0.7 |
| | KCenter | 75.6±0.5 | 82.7±0.5 | 84.8±0.7 | 85.2±0.4 | 57.3±0.9 | 70.9±0.7 | 76.1±0.7 | 78.0±0.4 | 40.8±0.4 | 48.6±0.6 | 53.1±1.0 | 54.7±0.7 |
| | Pr4Rel | 76.2±0.9 | 77.2±0.7 | 83.5±0.5 | 85.4±0.5 | 62.1±0.4 | 72.2±0.6 | 75.3±0.6 | 77.9±0.5 | 20.9±0.7 | 39.0±1.3 | 48.0±0.7 | 52.7±0.5 |
| | FDMat | 76.3±0.5 | 82.0±0.7 | 84.0±0.7 | 85.3±0.5 | 59.7±1.1 | 70.8±0.5 | 74.2±0.5 | 77.4±0.4 | 34.4±1.3 | 47.1±0.4 | 50.8±0.7 | 53.6±0.5 |
| | RoP | **83.1±0.9** | **87.3±0.4** | **88.0±0.4** | **86.9±0.3** | **81.8±0.6** | **83.4±0.4** | **79.5±0.3** | **78.5±0.3** | **46.6±0.9** | **52.7±0.6** | **55.1±0.4** | **55.2±0.4** |

extreme pruning. And the probability that these hard samples are noisy labels is very high, which makes it difficult for the model to learn these hard samples in the case of limited samples. On the contrary, some methods such as Small loss and Moderate consider covering easy samples, so that it is easy for the model to learn easy samples guaranteeing the performance of the model under extreme pruning. In Tab. 1, RoP means that samples with smaller NLI-Score are selected for pruning, while $RoP_B$ means that NLI-Score is sampled by using coverage. Due to NLI-Score considers the consistency between neighboring samples, directly using RoP to select easy samples can achieve excellent results on CIFAR-10N (Worst) and CIFAR-100N. $RoP_B$ further considers the coverage of hard and easy samples, and weighs the performance under different pruning rates and ensure the robustness.

Next, in Tab. 2, we analyze the performances of data pruning methods without using re-labeling method. Our findings reveal that the RoP method significantly outperforms the other baselines, particularly in the CIFAR-10N (Worst) dataset. This superior performance can be attributed to the NLI-Score metric, which evaluates label consistency among neighboring samples, thereby enabling RoP to select the maximum number of clean samples possible.

To further validate the efficacy of RoP in selecting clean samples using the LNI-Score, we compare the number of noisy samples contained across different methods, as presented in Tab. 3. The results indicate that the subset selected by RoP contains only 4.8% noisy labels, in stark contrast to 17% for other methods when 20% of the samples from CIFAR-10N are selected. This demonstrates that the NLI-Score effectively distinguish clean samples from noise label dataset.

Then, in Fig. 4, we compare the performances on a larger real noisy dataset, WebVision. We observe that methods considering only single-sample information, such as Small and GraNd, fail on large-scale datasets. However, the methods considering neighboring samples such as Pr4ReL and

Table 3: Ratio of noisy samples in selected subset.(10K images are selected from CIFAR-10N.)

| Re-label Model | Selection Methods | CIFAR-10N(Random) | |
| --- | --- | --- | --- |
| | | Test Acc. | %Noisy |
| SOP+ | Uniform | 87.5 | 17.8 |
| | KCenter | 86.3 | 17.0 |
| | Forget | 82.4 | 17.0 |
| | Pr4ReL | 88.1 | 17.0 |
| | RoP | **89.2** | **4.8** |

Table 4: Comparison of baselines and RoP on ImageNet-N. (Noise ratio≈20%)

| Re-label Model | Selection Methods | ImageNet-1K | | | |
| --- | --- | --- | --- | --- | --- |
| | | 0.05 | 0.1 | 0.2 | 0.4 |
| SOP+ | Uniform | 27.8 | 42.5 | 52.7 | 59.2 |
| | Small | 22.8 | 31.4 | 42.7 | 54.4 |
| | Forget | 4.1 | 8.3 | 50.6 | 57.2 |
| | Pr4ReL | 30.2 | 44.3 | 53.5 | 60.0 |
| | RoP | **31.0** | **44.7** | **55.6** | **63.4** |

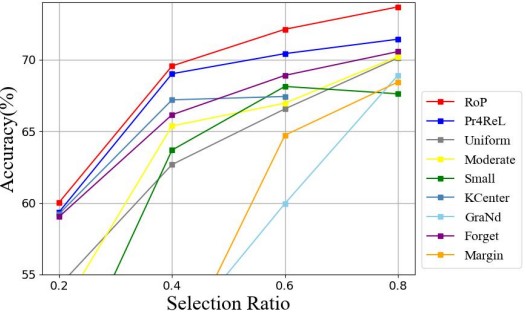

Figure 4: Comparison of data pruning methods on the large-scale real noisy WebVison dataset.

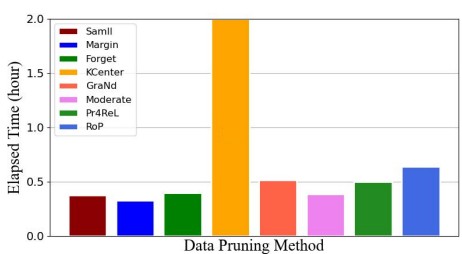

Figure 5: Data pruning efficiency on ImageNet-N with a selection ratio of 0.2.

KCenter still have certain robustness. In contrast, methods that leverage neighboring samples, like Pr4ReL and KCenter, demonstrate greater robustness. Importantly, RoP consistently outperforms the compared methods on the WebVison dataset, especially as the subset size increases.

### 4.5 EXPERIMENTS ON SYNTHESIS NOISY DATASET.

In Tab. 4, we further validate the performance of RoP on synthesis noisy dataset ImageNet-N, which incorporates 20% synthetic label noise into the widely used ImageNet-1K dataset. Our findings indicate that RoP consistently outperforms random selection and other data pruning methods across various pruning rates, with its advantages becoming most pronounced at lower pruning rates.

### 4.6 SELECTION EFFICIENCY

In data pruning tasks, the efficiency of sample selection is crucially important. In Fig. 5, we provide a detailed comparison of selection efficiency across various methods. Our analysis reveals that KCenter is ill-suited for large-scale datasets, primarily due to the substantial computational costs associated with sub-modular functions. In contrast, the other methods exhibit only minor differences in selection time, with the efficiency of RoP closely paralleling that of GraNd, both of which operate within an acceptable range.

### 4.7 DIFFERENT RE-LABELING METHODS

In Tab. 5, we investigate the impact of various re-labeling methods on our proposed two-stage framework. Specifically, we employ DivMix Li et al. (2020) as the re-labeling technique and conduct experiments with different pruning rates on the CIFAR-N (Worst) dataset. Our results reveal that methods utilizing DivMix exhibit a decline in performance compared to those using SOP+ Liu et al. (2022b) . Nevertheless, our proposed approach maintains superior performance relative to the other methods, even when DivMix is employed as the re-labeling strategy.

Table 5: Comparison of baselines and RoP by different re-labeling methods on CIFAR-10N (Worst). All methods use RreAct ResNet-18 with 3 different random seeds. The best results are in bold.

| Re-label Methods | Selection Methods | CIFAR-10 Worst | | | |
|---|---|---|---|---|---|
| | | 0.2 | 0.4 | 0.6 | 0.8 |
| DivMix | Random | $83.2\pm0.2$ | $88.5\pm0.1$ | $90.2\pm0.0$ | $91.4\pm0.0$ |
| | Small | $70.3\pm0.6$ | $80.3\pm0.2$ | $89.1\pm0.0$ | $92.1\pm0.1$ |
| | Margin | $61.3\pm0.8$ | $75.1\pm0.7$ | $85.3\pm0.2$ | $90.2\pm0.1$ |
| | KCenter | $82.7\pm0.8$ | $88.4\pm0.1$ | $90.6\pm0.1$ | $92.2\pm0.0$ |
| | Forget | $78.3\pm0.6$ | $88.3\pm0.2$ | $90.4\pm0.1$ | $92.0\pm0.2$ |
| | GraNd | $18.5\pm1.7$ | $25.5\pm0.9$ | $49.3\pm0.9$ | $88.0\pm0.5$ |
| | SSP | $81.4\pm2.5$ | $86.5\pm1.9$ | $89.6\pm1.2$ | $91.9\pm0.4$ |
| | Moderate | $81.4\pm1.2$ | $86.5\pm0.6$ | $90.0\pm0.6$ | $91.6\pm0.2$ |
| | Pr4ReL | $83.7\pm0.4$ | $88.6\pm0.4$ | $90.8\pm0.2$ | $92.4\pm0.2$ |
| | FDMat | $82.3\pm1.7$ | $88.1\pm0.4$ | $91.2\pm0.3$ | $92.3\pm0.2$ |
| | RoP | $83.4\pm0.3$ | $\mathbf{89.3}\pm\mathbf{0.3}$ | $91.6\pm0.3$ | $92.4\pm0.2$ |
| | RoP$_B$ | $\mathbf{84.4}\pm\mathbf{0.1}$ | $88.6\pm0.2$ | $\mathbf{92.4}\pm\mathbf{0.3}$ | $\mathbf{93.2}\pm\mathbf{0.2}$ |

Table 6: Ablation study of the two stages in RoP. No-Rec. means neither feature nor label propagation is used to rectify neighborhood labels. Rec. means using both feature and label propagation.

| No-Rec. | Rec. | Re-labeling | CIFAR-10 (Worst) | | | |
|---|---|---|---|---|---|---|
| | | | 0.2 | 0.4 | 0.6 | 0.8 |
| ✓ | | | 80.5 | 82.1 | 77.8 | 77.8 |
| | ✓ | | 81.8 | 83.4 | 78.5 | 78.5 |
| ✓ | | ✓ | 85.4 | 90.2 | 92.0 | 92.9 |
| | ✓ | ✓ | **86.4** | **90.8** | **92.4** | **93.3** |

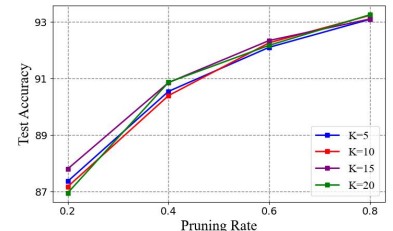

Figure 6: Impact of neighborhood sample size $K$ on CIFAR-10N (Worst).

## 4.8 ABLATION STUDIES

In Tab. 6, we conduct ablation experiments to evaluate the effectiveness of the proposed two-stage method. First, we examine the role of feature propagation and label propagation in rectifying the predictions of neighboring samples. "Rec." denotes the method that employs both feature propagation (FP) and label propagation (LP) techniques to rectify the predictions of neighboring samples, while "No-Rec" refers to the method that dose not utilize these techniques. As shown in Tab. 6, after using FP and LP, the performance is improved compared to directly calculating the neighboring label inconsistency even without using the re-labeling method. The results show that neighboring label correction is effective and relabeling is crucial for noisy label processing. However, not using the re-labeling method will cause the model to overfit to the noisy labels at low pruning rates, resulting in a decrease in model performance. When the re-labeling method is used, the model avoids overfitting to noisy labels to some extent, and the performance does not degrade at low pruning rates.

In addition, Fig. 6 presents ablation studies on the number of neighboring samples $K$ used to correct neighboring prediction when obtaining NLI-Score. We find that a smaller value $K$ leads to a decreased performance when the selection rate is low. However, as the number of selected samples increases, the influence of $K$ tends to be stabilized. When K is in the range of 10-15, the subset can achieve reasonable performance.

## 5 CONCLUSION

We present a two-stage robust data pruning method, called RoP, designed for noisy label scenarios. Initially, RoP identifies the clean samples and subsequently re-labels these selected samples. To identify clean samples, RoP introduces a novel metric termed the neighborhood label inconsistency score (NLI-Score), which quantifies the divergence discrepancy between a given label and predictions of neighboring samples. During the process of obtaining NLI-Score, RoP employs feature and label propagation to rectify neighboring predictions, thereby exploring the interrelations among them more deeply. Then, RoP selects samples by NLI-Score, and uses the density-based coverage sampling method to balance the number of easy and hard samples, which ensures the robustness across different pruning rates. Finally, RoP re-labels the selected subset using different re-labeling methods. RoP is limited by the performance of the chosen re-labeling methods. Extensive experiments demonstrate the effectiveness of the proposed data pruning method under noisy label scenarios.

## 6 REPRODUCIBILITY STATEMENT

The code is available at anonymous link $https : //anonymous.4open.science/r/RoP - 6D60$.

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

# A    APPENDIX

---

**Algorithm 2** CCS: Coverage-centric Coreset Selection

---

**Input**: $\mathbb{S} = \{\text{NLI-Score}(x_i)\}_{i=1}^n$: dataset with the NLI-Score for each example;
$\alpha$: data pruning rate;
$\beta$: hard cutoff rate ($\beta \leq 1 - \alpha$); $k$: the number of strata.
**Output**: Pruned data $\mathcal{S}^*$

1: $\mathbb{S}' \leftarrow \mathbb{S} \backslash \{[n * \beta] hardest examples\}$ ;
2: $\mathbb{B}' \leftarrow \{\mathbb{B}_i \backslash \{[n * \beta] hardest examples\}$ ;
3: $R_1, R_2, ... R_k \leftarrow$ Split scores in $\mathbb{S}'$ into $k$ ranges with an even range width ;
4: $\mathcal{B} \leftarrow \{\mathbb{B}_i, : \mathbb{B}_i :$ consists of examples whose scores are in $R_i, i = 1...k\}$;
5: **while** $B \leq \varnothing$ **do**
6:    $R_1, R_2, ... R_k \leftarrow$ Split scores in $\mathbb{S}'$ into $k$ ranges with an even range width ;
7:    $\mathbb{B}_{min} \leftarrow \underset{\mathbb{B} \in B}{\arg\min} |\mathbb{B}|$;
8:    $m_B \leftarrow \min\{|\mathbb{B}_{min}|, |\frac{m}{|B|}|\}$;
9:    $\mathbb{S}_B \leftarrow$ randomly sample $m_B$ examples from $\mathbb{B}_{min}$;
10:   $\mathbb{S}_c \leftarrow \mathbb{S}_c \cup \mathbb{S}_B$
11:   $B \leftarrow B \varnothing \{\mathbb{B}_{min}\}$
12:   $m \leftarrow m - m_B$
13: **end while**
14: **Return** $\mathbb{S}^*$

---

## A.1    DETAILS FOR COVERAGE SAMPLING

We leverage the density-based Coverage- centric Coreset Selection (CCS) Zheng et al. (2022b) to trade off the number of hard and easy samples, as outlined in Algorithm 2. CCS first partitions the dataset into distinct, non-overlapping strata, with each stratum defined by a fixed-length range of NLI-Scores. Though the NLI-Score ranges are uniform across strata, the number of examples within each stratum may vary. CCS then sets an initial budget on the number of examples to be selected from each stratum, based on the desired pruning rate. However, if a particular stratum contains fewer examples than the allocated budget, the excess budget is evenly redistributed across the remaining strata.

## A.2    DETAILS FOR SYNTHESIS IMAGENET-N

Since ImageNet-1K is a clean dataset with no known real label noise, we inject the synthetic label noise to construct ImageNet-N. Specifically, we inject asymmetric label noise to mimic real-world label noise following the prior noisy label literature. When targeting an $r\%$ noise ratio for ImageNet-N, we randomly select $r\%$ of the training examples from each class $c$ in ImageNet-1K and then systematically flip their labels to the next consecutive class $c + 1$, i.e., class 0 into class 1, class 1 into class 2, and so on. This deliberate label flipping strategy is reasonable, as consecutive classes are often semantically related, belonging to the same high-level conceptual category. For the selected examples from the final class 1000, we uniquely flip their labels to class 0, completing the circular noise injection process. This holistic label corruption approach serves to recreate the complex, heterogeneous noise characteristics typically encountered in real-world visual recognition datasets, providing a more realistic test environment for our subsequent research endeavors.

## A.3    LIMITATION AND SOCIAL IMPACT

### A.3.1    LIMITATION

While the RoP has consistently demonstrated its effectiveness in tackling classification tasks involving real-world and synthetically introduced label noise, its applicability on datasets plagued by open-set noise or containing out-of-distribution examples remains to be validated. Moreover, we have not yet assessed the efficacy of RoP when applied to state-of-the-art deep learning models,

Table 7: Summary of the hyperparameters for training SOP+ on the CIFAR-10N/100N,Webvision, and ImageNet-N datasets.

| | Hyperparamters | CIFAR-10N | CIFAR-100N | WebVision | ImageNet-N |
|---|---|---|---|---|---|
| | architecture | PreAct PresNet18 | PreAct PresNet18 | InceptionResNetV2 | ResNet50 |
| | warm-up epoch | 10 | 30 | 10 | 1 |
| Training | training epoch | 300 | 300 | 100 | 10 |
| Configuration | batch size | 128 | 128 | 32 | 32 |
| | learning rate(lr) | 0.02 | 0.02 | 0.02 | 0.02 |
| | lr scheduler | Cosine Annealing | Cosine Annealing | MultiStep-50th | MultiStep-50th |
| | $\lambda_C$ | 0.9 | 0.9 | 0.1 | 0 |
| | $\lambda_B$ | 0.1 | 0.1 | 0 | 0 |
| SOP+ | lr for $u$ | 10 | 1 | 0.1 | 0.1 |
| | lr for $v$ | 100 | 100 | 1 | 1 |

such as large language models and vision-language architectures. Verifying the performance of RoP across this expanded range of datasets and model paradigms would be immensely valuable, as the need for robust data pruning strategies in the face of annotation noise is a ubiquitous challenge permeating a wide spectrum of real-world applications. Additionally, the Robustness to Perturbations approach has yet to be validated in other realistic data pruning scenarios, such as continual learning and neural architecture search, where the selective retention of informative examples is of paramount importance. We intend to address these crucial research gaps in our future work. By rigorously evaluating the versatility and generalizability of RoP across diverse datasets, model architectures, and application domains, we can further solidify its standing as a powerful and adaptable tool for mitigating the detrimental effects of label noise.

### A.3.2 SOCIAL IMPACT

When it comes to preserving model performance while simultaneously reducing computational costs and energy consumption – which can lead to tangible benefits like lowering carbon dioxide emissions – we recognize the inherent challenges involved. However, we firmly believe that the techniques and approaches we explore in this work do not lend themselves to any nefarious or negative applications.

It is our conviction that by optimizing model performance and computational efficiency hand-in-hand, we can pave the way for wider adoption of AI technologies while minimizing their environmental footprint. This dual objective is a key driver behind our research, as we strive to create practical, ethical, and impactful solutions that benefit both the technical and the social realms. We remain steadfast in our commitment to responsible innovation, ensuring that our advancements in machine learning serve the greater good and do not give rise to any concerning social ramifications.

### A.4 EXPERIMENT DETAILS

In Tab. 6, we provide a comprehensive summary of the configurations and hyperparameters employed during the training of the Re-labeling stage. The hyperparameters for the SOP+ method have been favorably configured in accordance with the original publication Liu et al. (2022a). SOP+ involves several key hyperparameters: $\lambda_C$ for weighting the self-consistency loss, $\lambda_B$ for weighting the class-balance objective, and learning rates for training its additional variables $u$ and $v$. Specifically, for CIFAR-10N, we use $\lambda_C = 0.9$ and $\lambda_B = 0.1$, and set the learning rates of $u$ and $v$ to 10 and 100, respectively. For CIFAR-100N, the hyperparameters are set as $\lambda_C = 0.9$, $\lambda_B = 0.1$, the learning rates of $u$ and $v$ to 1 and 100, respectively. On the WebVision dataset, we employ $\lambda_C = 0.1$ and $\lambda_B = 0$, and the learning rates of $u$ and $v$ to 0.1 and 1, respectively. For the ImageNet-N dataset, the hyperparameters are $\lambda_C = 0$, $\lambda_B = 0$, and the learning rates of $u$ and $v$ are 0.1 and 1, respectively.

Furthermore, the hyperparameters of all compared data pruning methods are also favorably configured based on the recommendations from their respective prior works. Specifically, for CIFAR-10N and CIFAR-100N, A PreAct Resnet-18 is trained for 300 epochs using SGD with a momentum of 0.9, a weight decay of 0.0005, and a batch size of 128. The initial learning rate is 0.02, and it is

decayed with a cosine annealing scheduler. For WebVision, InceptionResNetV2 is trained for 100 epochs with a batch size of 32. For ImageNet-N, ResNet-50 model is trained for 50 epochs with a batch size of 64 and an initial learning rate of 0.02, also decayed with a cosine annealing scheduler. All methods are implemented with PyTorch 1.8.0 and executed on NVIDIA Tesla A100 GPUs.

