# OpenReview forum: "Noisy Data Pruning by Label Distribution Discrimination"
_ICLR.cc/2025/Conference — ICLR 2025 Conference Withdrawn Submission_

### Official Review · Reviewer_oDDr · 2024-10-28

**Soundness:** 1
**Presentation:** 2
**Contribution:** 1
**Rating:** 1
**Confidence:** 5

**Summary:**

This paper introduces a novel two-stage robust data pruning method (RoP) aimed at datasets with noisy labels. The first stage identifies clean samples using a Neighborhood Label Inconsistency Score (NLI-Score), followed by a second stage that re-labels the selected samples. RoP employs feature and label propagation to enhance the accuracy of neighboring predictions and uses density-based coverage sampling to balance the number of easy and hard samples across different pruning rates. Extensive experiments demonstrate the effectiveness of RoP on both synthetic noisy datasets and real-world benchmarks.

**Strengths:**

1. The paper is well-organized, with clear presentations of methodology, experiments, and conclusions that effectively guide the reader through the research.

2. The authors conduct extensive experiments across various datasets, including real-world noisy datasets and synthetic noise datasets, which helps to validate the robustness and applicability of the method.

**Weaknesses:**

1. I find this paper completely unoriginal; it merely applies existing techniques from the noisy label learning task to the domain of data pruning. The "FEATURE PROPAGATION" proposed by the authors is present in many works on noisy label learning [1-4]. The authors additionally utilize Equation 5, which involves fusing a sample's own features with those of surrounding samples to improve its own features; however, this may not be very meaningful. Many studies have shown that, in noisy label learning, while model predictions may be misled by noisy labels, the learned features tend to remain reliable. The "LABEL PROPAGATION" proposed by the authors is also found in many works on noisy label learning [5-6]. For the re-labeling part, the authors even explicitly mention using the existing state-of-the-art method, SOP+.

2. Why are methods related to noisy label learning not compared in the experiments? Many existing methods can be easily adapted to the scenarios presented in this paper.

[1] Multi-Objective Interpolation Training for Robustness to Label Noise. CVPR 2021

[2] Selective-Supervised Contrastive Learning with Noisy Labels. CVPR 2022

[3] RankMatch: Fostering Confidence and Consistency in Learning with Noisy Labels. ICCV 2023

[4] Learning with Neighbor Consistency for Noisy Labels. CVPR 2024

[5] Jo-SRC: A Contrastive Approach for Combating Noisy Labels. CVPR 2021

[6] UNICON: Combating Label Noise Through Uniform Selection and Contrastive Learning. CVPR 2022

**Questions:**

See weaknesses.

---

> ### Author Response · Authors · 2024-11-14
> **Noisy Data Pruning by Label Distribution Discrimination**
>
> Thank you for your **amazing** comments. Anyway, I will answer your questions carefully.
>
> **W1**: It appears that Reviewer oDDR may not have fully engaged our paper. We are not directly using existing technology. The only similarity between literature [1-4] and our method is the use of neighborhood samples to identify clean samples, which is a technique adopted by most methods in noisy label learning.
>
> However, our method significantly differs in its execution. Unlike [1-4], which focuses solely on finding neighborhood samples without considering higher-order relationships, we construct a neighborhood graph through feature propagation. This allows us to correct the predicted labels of neighborhood samples via label propagation, which is fundamentally distinct from [1-4]. In fact, the studies in [1-4] support the validity of our neighborhood labeling technique for handling noisy labels.
>
> Furthermore, the reviewers demonstrate significant misunderstandings regarding **feature propagation** and **label propagation**.
> Firstly, feature propagation aims to capture the higher-order relationships among neighborhood samples by constructing a graph. During this process, only the neighborhood samples are used to build the graph structure; thus, Equation 5 does not incorporate the features of the sample itself, as clearly outlined in the construction of the neighborhood graph in Equation 3.
>
> Secondly, the reviewer's interpretation of label propagation is also flawed. The purpose of label propagation is to correct the predicted labels of neighborhood samples, enhancing the accuracy of the distribution differences between the neighborhood sample and the given sample. Notably, literature [5-6] lacks any procedure for correcting neighborhood labels.
>
>
> To summarize, our approach is to deal with the data pruning problem in noisy label scenarios, which improves on the existing techniques. The literature [1-4] listed by the reviewer has proved the rationality of using neighborhood samples to find noisy labels, and the literature [5-6] has highlighted the innovativeness of using label propagation to correct neighborhood labels in our method.
>
> **W2**:  Noisy label learning methods [1-6] cannot be directly applied to data pruning tasks, and these methods change the loss function and destroy the fairness of data pruning evaluation. In addition, in order to ensure fairness, the existing methods such as Pr4ReL[7] and FDMat[8] are not compared with these methods in the data pruning community.
>
> [7] Robust Data Pruning under Label Noise via Maximizing Re-labeling Accuracy. NeurIPS 2023
>
> [8] Feature Distribution Matching by Optimal Transport for Effective and Robust Coreset Selectio. AAAI 2024

---

> ### Comment · Reviewer_oDDr · 2024-11-14
> **Thank you for your response, although it has already been removed.**
>
> First of all, I indeed made a mistake. [5] and [6] correspond to NEIGHBORHOOD LABEL INCONSISTENCY ESTIMATE, not LABEL PROPAGATION.
>
> Secondly, as a researcher in noisy label learning, my emphasis is on 'it merely applies existing techniques from the noisy label learning task to the domain of data pruning'. Equations 1-3 and 7-8 correspond to techniques that are very common in the noisy label learning field, while SOP+ is an existing state-of-the-art noisy label method. What is your unique contribution? Is it sufficient to match the standards of ICLR?
>
> Thirdly, my emphasis is on 'Many existing methods can be easily adapted to the scenarios presented in this paper'. I understand that these methods cannot be directly applied.
>
> Finally, many research areas overlap, and it is necessary for a new domain to be compared with methods from related fields. Data pruning overlaps with sample selection strategies in active learning from the perspective of data selection, and the noisy label setting in this paper is also closely related to the noisy label learning scenario. Therefore, I believe the authors should be familiar with the technical contributions from related fields, as conducting repetitive research serves no meaningful purpose.

---

> ### Author Response · Authors · 2024-11-14
> **Noisy Data Pruning by Label Distribution Discrimination**
>
> Thanks for your comments.
>
> First of all, thank you for admitting your mistakes. Because where you admit to misunderstanding is our contribution. You completely ignore Equations 4-6 and keep emphasizing that it is work duplication as long as the techniques approved by references [1-4] are adopted.
>
> Second, where we differ from the previous approach is in Eqs. 4-6, and elsewhere we adopt similar techniques such as Eqs. 1-3 and 7-8. We have responded to the differences in detail before.
>
> Finally, this is not a repeated study, we apply the appropriate techniques to the corresponding scenarios. Such work is required to deal with noisy data pruning for real-world scenarios.

---

### Official Review · Reviewer_g36C · 2024-11-03

**Soundness:** 2
**Presentation:** 1
**Contribution:** 2
**Rating:** 3
**Confidence:** 3

**Summary:**

This paper studies the problem of noisy data pruning which aims to prune noisy large-scale datasets into concise subsets. The authors first reveal that geometry-based methods exhibit greater resilience to noisy labels compared to gradient-based methods. Then, a discrimination, pruning, and re-labeling method is proposed to conduct noisy data pruning. Specifically, noisy label discrimination is achieved by neighborhood label inconsistency estimation, after feature and label propagation. Then, the pruned set is selected by ensuring coverage on both easy and hard samples. Finally, re-labeling is achieved by SOTA noisy label learning methods. Experiments show the effectiveness of the proposed method.

**Strengths:**

- This paper addresses the issue of noisy data pruning, which is crucial in real-world applications.
- The proposed method achieves SOTA results against existing baselines.

**Weaknesses:**

- The presentation of this paper is poor according to the following aspects:
  - Lacking in-depth analysis of the difference between the proposed method and previous SOTA Pr4ReL, since Pr4ReL also follows a selection and relabeling paradigm and uses neighborhood information. It is important to explain the superiority of the proposed method. Also, authors are encouraged to add noisy data pruning methods in the section of related work.
  - The motivation for feature propagation and label propagation is unclear. Besides, what if directly applying label propagation without feature propagation?
- Line 237 mentions that a model is required to be trained on the noisy label datasets. Since the purpose of dataset pruning is to reduce the training cost, is it reasonable and fair to access the trained model? If so, please refer to some related works.
- The process of pruning and re-labeling actually follows existing works, i.e., a SOTA noisy label learning method SOP+ and Coverage Coreset Sampling strategy, limiting the novelty of the proposed method.
- Typo: In line 309 "retaining retaining"

**Questions:**

In line 250, how to get the ground-truth label of the sample?

---

> ### Author Response · Authors · 2024-11-14
> **Noisy Data Pruning by Label Distribution Discrimination**
>
> Thank you very much for your valuable comments. We will address each question with careful consideration and comprehensive detail.
>
> **W1**: (1) We will improve our analysis of  Pr4Rel and the related work of noise data pruning methods. The primary distinction between our approach and Pr4Rel is that we focus on identifying the most likely clean samples, while Pr4Rel selects easily labelable samples through empirical analysis. Our method demonstrates superior performance, as shown in Table 2, particularly when the re-labeling method is not employed.
>
> (2) The motivation for feature and label propagation is to accurately identify clean samples. To do this, we calculate the distribution difference between the predicted labels of neighborhood samples and the true label of the given sample. Since the predicted labels of neighborhood samples affect the selected of clean samples, we correct them through graph construction, a step not taken by previous methods. Feature and label propagation are interconnected; label propagation relies on the relationships established in feature propagation. Thus, label propagation cannot be directly applied without using feature propagation.
>
> **W2**: Our method follows the standard experimental setup in the data pruning community, where we first train on the full dataset for 10 epochs to obtain a pre-trained model. This model is then used for data pruning, and the selected data is employed to train the final model to reduce cost. Following the experimental conditions of Pr4Rel [1] and FDMat [3].
>
> **W3**: Many current methods in the dataset pruning community utilize re-labeling and coverage coreset sampling (CCS) strategies directly. For instance, Pr4R2L [1] employs re-labeling, and MB [2] uses CCS. Our contribution is primarily in the selection of clean samples through feature and label propagation, and the two-stage architecture design for noisy data pruning.
>
> **W4**: We will correct these details and remove the duplicate word 'retaining'.
>
> **Q1**: Since the dataset is already labeled, the true label of a sample is known.
>
> [1] Robust Data Pruning under Label Noise via Maximizing Re-labeling Accuracy. NeurIPS 2023
>
> [2] Mind the Boundary: Coreset Selection via Reconstructing the Decision Boundary. ICML 2024.
>
> [3] Feature Distribution Matching by Optimal Transport for Effective and Robust Coreset Selectio. AAAI 2024

---

### Official Review · Reviewer_Y5LY · 2024-11-04

**Soundness:** 2
**Presentation:** 2
**Contribution:** 3
**Rating:** 3
**Confidence:** 4

**Summary:**

In this paper, the authors propose a novel geometry-based noisy data pruning method. It consists of two stages and uses feature propagation and label propagation for reliable neighboring predictions. Experiments demonstrated quite good results.

**Strengths:**

- It proposes a novel two-stage noisy data pruning method.
- It employs novel feature propagation and label propagation to refine neighboring predictions.

**Weaknesses:**

- The motivation is weak. First, the loss-based method GraNd is not designed and optimized, especially for noisy samples. Thus it is not suitable to choose GradNd to represent the method. Second, it is not convincing to get the conclusion that the geometry-based approach is better than the loss-based approach by simply comparing the two methods. More methods and more noise settings are needed. Moreover, the statement in line 68 applies to all methods, and also can’t lead to the conclusion.
- I recommend the discussion about the noisy data pruning method and sample selection method. It is an important topic that when the pruning rate is limited, the approximation of the noise rate is neglected. Therefore, the result is destined to be sub-optimal.
- From Table 1, Table 2 and Table 6, it seems that the relabeling method is more dominant than selection. I slightly question the motivation to use pruning for mitigating noisy samples since relabeling has solved this problem well.
- The presentation needs to be polished, e.g. the suddenly appeared GraNd in line 52, and some typos, such as 87.9±-1.3 in Table 1 and brackets in line 6.
- The types of noise are insufficient. Symmetric label noise should be taken into experiments.

**Questions:**

- It’s hard to understand the so-called “intuitive method” in line 70: Since the clean samples are found in the first stage, why do you need to relabel them?
- Are the gradient-based method in line 15 and the loss-based method in line 51 the same thing?

---

### Official Review · Reviewer_Wddu · 2024-11-04

**Soundness:** 2
**Presentation:** 2
**Contribution:** 2
**Rating:** 5
**Confidence:** 3

**Summary:**

This paper studied a task combination of label noise and core-set selection. The proposed method, RoP, is a graph-like sample selection strategy that integrates feature propagation and label propagation. The experiment is extensive.

**Strengths:**

- Introduces an innovative NLI-Score for identifying noisy samples by leveraging neighboring sample consistency.
- Combines feature and label propagation effectively to reduce selection bias in noisy data scenarios.

**Weaknesses:**

- The discussion on a closely related work, Pr4ReL, requires expansion. (1) Pr4ReL also employs a neighborhood-based strategy for selection and relabeling. Further insights into why RoP outperforms Pr4ReL, along with a more detailed comparison, would strengthen the analysis. (2) Additionally, RoP's performance on CIFAR-10N/100N, as shown in Table 1, does not surpass that of Pr4ReL, which merits further examination.

-  The authors employ CCS for coverage pruning; however, the adaptation of CCS to NLI-Scores is not clearly explained. Additional clarification would help readers understand this approach.

- Experimentation. (1) Additional experimentation on large-scale datasets is recommended, as Mini-WebVision contains only 66k training images. An experiment on Clothing-1M, comparable to the setup in Pr4ReL, would provide more robust validation. (2) The ablation study is limited. In Table 6, the authors should include separate analyses of using feature and label propagation independently.

- Minor typos, such as "retaining" on Line 309, should be corrected for clarity.

**Questions:**

see weaknesses.

---

### Note · Authors · 2024-11-15

**Comment:**

Due to one reviewer's misunderstanding and blindly giving a high confidence score, it is difficult for me to accept, so I plan to withdraw the manuscript. However, I hope that for other papers, AC can exclude some malicious reviewers, especially reviewers who blindly give high confidence without reading the paper.

**Withdrawal Confirmation:**

I have read and agree with the venue's withdrawal policy on behalf of myself and my co-authors.